# The Role of Pannexin-1 Channels in HIV and NeuroHIV Pathogenesis

**DOI:** 10.3390/cells11142245

**Published:** 2022-07-20

**Authors:** Cristian A. Hernandez, Eugenin Eliseo

**Affiliations:** Department of Neuroscience, Cell Biology, and Anatomy, University of Texas Medical Branch (UTMB), Galveston, TX 77555, USA; craherna@utmb.edu

**Keywords:** dementia, biomarker, purinergic, comorbidities, cure

## Abstract

The human immunodeficiency virus-1 (HIV) enters the brain shortly after infection, leading to long-term neurological complications in half of the HIV-infected population, even in the current anti-retroviral therapy (ART) era. Despite decades of research, no biomarkers can objectively measure and, more importantly, predict the onset of HIV-associated neurocognitive disorders. Several biomarkers have been proposed; however, most of them only reflect late events of neuronal damage. Our laboratory recently identified that ATP and PGE_2_, inflammatory molecules released through Pannexin-1 channels, are elevated in the serum of HIV-infected individuals compared to uninfected individuals and other inflammatory diseases. More importantly, high circulating ATP levels, but not PGE_2_, can predict a decline in cognition, suggesting that HIV-infected individuals have impaired ATP metabolism and associated signaling. We identified that Pannexin-1 channel opening contributes to the high serological ATP levels, and ATP in the circulation could be used as a biomarker of HIV-associated cognitive impairment. In addition, we believe that ATP is a major contributor to chronic inflammation in the HIV-infected population, even in the anti-retroviral era. Here, we discuss the mechanisms associated with Pannexin-1 channel opening within the circulation, as well as within the resident viral reservoirs, ATP dysregulation, and cognitive disease observed in the HIV-infected population.

## 1. Introduction

Human immunodeficiency virus type 1 (HIV), the causative agent for acquired immunodeficiency syndrome (AIDS), was identified four decades ago [1]. Despite the introduction of several successful anti-retroviral regimens to reduce systemic replication, HIV reservoirs seed early after infection, preventing viral eradication [2,3]. Upon anti-retroviral therapy (ART) interruption, the virus in latently infected cells rebounds, repopulating the individual with the virus in a few weeks [4,5,6,7]. Despite ART reducing systemic viral replication, chronic events such as residual viral protein expression and secretion are ongoing, inducing chronic neurotoxicity and associated cognitive decline [8,9,10].

The main objective of the ART regimens is to reduce viral replication and allow the body to reconstitute the immune system. An additional goal of ART is to target infected cells to enable them to present viral antigens to be recognized by the immune system and kill them [11,12]. However, most mechanisms of viral eradication require active replication, which is the basis for the current “shock and kill” approaches [13]. The “shock and kill” approach relies on the intrinsic toxicity of viral proteins in productively infected cells and a better recognition of the infected cells by the immune system, reducing the reservoir pool [14]. However, this approach has several issues, including the potential uncontrolled virus reactivation and associated immune compromise [15]. Moreover, most latently infected cells are not “uniform”; they cannot or do not respond to a single reactivator agent such as histone deacetylases or protein kinase C activators, suggesting different mechanisms of silencing and reactivation [16,17,18,19]. In addition, the reactivation mechanism depends on the cell type infected, including different cell populations, differentiation stage, proliferation, tissue compartmentalization, and cell lineage. Thus, all viral reservoirs need a specific mechanism of reactivation to be eliminated [13,20,21]. These issues complicate not only viral eradication, but also provide a unique viral evolution that can better adapt into particular immune and ART situations, increasing the flexibility of the virus to persist for extended periods [22]. Only recently, experiments in humans with interrupted ART indicate that viral rebound is diverse and cannot only be explained by the circulating virus, suggesting a significant contribution of multiple viral reservoirs to viral rebound and the viral repopulation of the body [23]. Another complication of the current cure strategies is the clear identity of all or most of the viral reservoirs in humans and animal models. Currently, the identification, quantification, and characterization of viral reservoirs are mostly focused on the blood, which probably does not represent the real viral reservoirs present in different tissues, including the brain [24,25]. The most thoroughly characterized component of the viral reservoirs is the nature of transcriptionally silent DNA proviruses integrated within memory CD4^+^ T lymphocytes [26,27]. However, several questions remain open to better understand these few latently infected cells, including the integration sites, mechanism of viral silencing, survival, reactivation, viral spread, and the host factors behind all of these mechanisms. In addition, some cell populations, such as central and stem cell memory T cells, could be cycling or exhibit self-renewal without activation or viral replication [28,29]. In addition, rare CD4^+^ T lymphocytes, such as effector memory T lymphocytes, have higher intact and inducible provirus frequencies, suggesting that particular cell populations can be better infected and reactivated [30]. However, our data in macrophage/microglia and astrocytes indicate that good infectivity and replication do not account for a good reservoir. For example, astrocytes have low infectivity and replication, but they have a highly efficient mechanism of viral transfer and amplification mediated by cell-to-cell contact not described in other reservoirs that spread infection by the soluble virus [31].

Additionally, in the context of HIV, these reservoirs are known to have dysregulation of hemichannels and gap junction proteins, such as Pannexin-1 (Panx-1) and Connexin 43, which promote the local secretion of inflammatory molecules such as ATP (adenosine-5’-triphosphate) and lipids [32,33]. The secretion or propagation of these inflammatory molecules through these hemichannels or gap junctions can lead to bystander apoptosis [34]. Some mechanisms, including viral transfer, are shown in elite and exceptional elite controllers, where ongoing viral replication still occurs at a low level, but the body is capable of maintaining immunity and suppressing viral replication even in the absence of ART [35]. In these populations, the majority of proviruses are in a silent state; however, like in astrocytes, HIV controllers are highly heterogeneous, and many silencing mechanisms have not yet been characterized, including survival, latency, and immune evasion in the absence of ART, and bystander damage.

In addition, recent publications indicated that some CD4^+^ T cell populations in chronic HIV-infected individuals under ART are anergic and/or exhausted with specific markers such as immune checkpoint receptors programmed death protein 1 (PD-1), cytotoxic lymphocyte-associated protein 4 (CTLA-4), and others [36,37]. These mechanisms have not been examined in other viral reservoirs such as myeloid or astrocytes; however, these cell types are “embedded” in the tissues and are extremely difficult to examine compared to the circulating viral reservoirs present in the blood.

Interestingly, even in HIV-infected individuals on effective suppressive ART, cells with active replication and viral protein translation have been described [9,10,38]. Despite the expression of viral mRNA and proteins as potential identifiers of infected cells, HIV-infected cells persist for years to decades, suggesting alternative survival mechanisms or a slow turnover to maintain a pool of reservoirs in each tissue. As current HIV curative treatments, such as “shock and kill”, cannot effectively reactivate latent HIV due to the diffuse and heterogeneous nature of viral reservoirs, we must treat HIV infection as a chronic disease. Thus, the early diagnosis or identification of associated comorbidities as well as the mechanisms of toxicity can provide interventional strategies to prevent and even revert some long-term consequences of chronic HIV infection. This review is intended to detail current topics related to HIV, including persistence within viral reservoirs, biomarkers of infection, and the newly identified host protein, Panx-1, and its function as a mediator of potential biomarkers of CNS damage in the current ART era.

## 2. Viral Reservoirs in the Brain

The brain and the peripheral nervous system are tissues colonized by myeloid and astrocytic viral reservoirs early after primary infection. The brain lacks T cells and other immune components such as immunoglobulins due to the selectivity of the blood–brain barrier (BBB) that excludes most immune components [39]. However, the brain is susceptible to several neurotropic viruses, including polyomavirus JC, Zika virus, dengue, Japanese encephalitis virus, West Nile, and HIV [40,41,42,43,44,45]. In the brain, the main infected cells correspond to microglia/macrophages and a small population of astrocytes [31,46]. However, the role of myeloid and glial cells has only recently been examined. In the ART era, a wide spectrum of neurological diseases termed HIV-associated neurocognitive disorders (HAND) have been detected in about half of the HIV-infected population [47]. The mechanisms maintaining high prevalence of HAND in the current ART era are unknown [48]. Instead, several groups propose that viral reservoirs in microglia/macrophages and astrocytes play a key role in chronic central nervous system (CNS) dysfunction. However, only recently have significant efforts occurred to identify and characterize the CNS viral reservoirs but have resulted in surprising and unique data sets compared to latently infected peripheral T cells [49,50,51,52,53,54,55]. Thus, viral reservoirs in the brain are different and unique.

### 2.1. Microglia/Macrophages

Microglia are the resident macrophages of the CNS and perivascular macrophages as the major phagocytes in the brain [56]. However, astrocytes can also have limited immune antigen presentation, but significantly contribute to inflammation [57]. In the pre-ART era, microglia/macrophages were the predominant cells involved in inflammation and neuronal/glial apoptosis [58]. These pre-ART conditions were recapitulated in felines and macaques using or analyzing infected macrophages [59,60]. Most CNS damage was associated with excessive viral replication, inflammation, and secretion of neurotoxic proteins, resulting in neuronal compromise and changes in cognition [61]. Further, the damage was amplified by the recruitment of uninfected cells into areas with HIV [62], including the transmigration of uninfected and HIV-infected monocytes into the CNS [63]. Both microglia and macrophages harbor the virus and resist apoptosis, serving as viral reservoirs. Our data indicate that a small population of latently infected microglia/macrophages survives the infection and silences the virus by a mechanism that prevents the proper formation of the apoptosome, preventing cell death [64]. We also identified that survived latently infected microglia/macrophages had a unique metabolic profile similar to the metabolic profile observed in glioblastoma stem cells [53]. Therefore, microglia and brain macrophages are critical targets of the HIV reservoirs and must be included in future cure strategy approaches.

### 2.2. Astrocytes

Astrocytes are the most abundant cell population in the brain, and they control key synaptic and immune functions during normal and pathological functions [65]. Astrocytes are diverse, heterogeneous, and have area-specific functions. Examples include tanycytes, radial, Bergmann glia, protoplasmic, fibrous, velate, marginal glia, and perivascular ependymal glia [66]. Astrocytes have been described to be targeted by HIV in vitro and in vivo for several groups; however, the number of infected astrocytes is low (around 5%) [32,67,68]. Astrocytes are negative for CD4, but express CC Motif Chemokine Receptor 5 (CCR5) and CXC Motif Chemokine Receptor 4 (CXCR4), suggesting some entry restrictions [69]. Only recently, the improvement of several techniques to identify and characterize viral reservoirs in blood and tissues confirmed that astrocytes had low numbers of infected cells but more than sufficient to repopulate an entire animal in the absence of other infected cells [67]. Our studies using human astrocytes identified that despite the low rate of infection and low/undetectable replication, HIV-infected astrocytes could generate aberrant intracellular toxic signals such as inositol triphosphate (IP_3_) and calcium but did not result in apoptosis due to the presence of the nef protein preventing IP_3_ receptor overactivation and the subsequent apoptosome formation [70]. More importantly, residual viral replication is silenced even in the absence of ART, suggesting that astrocytes, but not other cell types, had intrinsic silencing mechanisms. However, upon viral reactivating agents, the few infected astrocytes can efficiently transfer the infectious virions to macrophages and T lymphocytes without astrocyte death, supporting astrocytes as a different viral reservoir. In addition, we demonstrated that most of the survival and viral silencing/reactivation mechanisms were mediated by the loss of proper interactions among the endoplasmic reticulum, the Golgi apparatus, and the mitochondria, all essential components of the apoptotic process [34,70]. Therefore, HIV-infected astrocytes correspond to a versatile and different kind of viral reservoir than lymphoid and myeloid, suggesting that new mechanisms of survival, escape, and viral silencing are present in these cells.

## 3. Crusade to Find Reliable Biomarkers to Identify Chronic and Acute HIV-Mediated Damage

A critical role of viral reservoirs in the current ART era is chronic bystander damage described in several tissues, including the CNS [70,71,72,73,74,75]. However, the mechanisms of damage or identifying specific disease biomarkers in the current ART era are lacking. Overall, biomarkers reflect or predict the changes in pathological conditions. Even then, in 50% of HIV-infected individuals, the field fails to identify the mechanisms of chronic CNS disease in the HIV-infected population. Several biomarkers of HAND have been proposed over the years, including neurofilament light chain (NFL) [71], β-Amyloid_1-42_ [76,77], calcium-binding protein B [75,78], extracellular vesicles [75,79], and the Wnt-related proteins [80]. Other groups identified immune markers such as soluble CD14 [81], soluble CD163 [82,83,84], neopterin [85,86], Cathepsin B [87], kynurenine to tryptophan ratio [88,89,90,91], monocyte chemoattractant protein-1 (MCP-1 or CCL2) [92,93,94], tumor necrosis factor-alpha [8,84,95], interleukin-6 (IL-6) [94,95], interferon-γ-inducible protein (IP-10 or CXCL-10) [94,96], interleukin-8 (IL-8 or CXCL-8) [96,97], interferon alpha [90,98,99], intercellular adhesion molecule-5 [100], lipopolysaccharide (LPS) [81,101], brain-derived neurotrophic factor (BDNF) [102], and several growth factors [103]. Despite the long list of potential biomarkers of HIV CNS disease, most are associated with the late events of tissue destruction or significant immune activation and cannot detect early or chronic stages of CNS compromise.

Other approaches to identifying the early stages of CNS damage include non-invasive neuroimaging methods. Although these methods are not designed to identify biomarkers, they could provide essential information on brain behavior and structural and metabolic compromise. Moreover, the type of brain damage induced by HIV infection has been examined using multiple techniques, including postmortem tissues, cognitive assessment, and imaging technologies such as diffusion basis spectral imaging (DBSI). Most structural approaches consistently show a decrease in gray matter volumes and white matter microstructural abnormalities, called hyper-densities, compared to uninfected individuals [104,105]. The hyper-densities have been associated with inflammation and compromised blood vessels or mini “stroke” areas based on cellularity [106,107,108,109]. However, the nature and mechanism of the damage were unknown. Our group demonstrated that the serum adenosine triphosphate (ATP) and prostaglandin E_2_ (PGE_2_) levels were elevated in HIV patients relative to samples obtained from uninfected individuals [110]. However, only ATP was predictive of the early stages of cognitive decline in the HIV-infected population [110]. We identified that viral reservoirs, myeloid and astrocytic, can release ATP, contributing to the purinergic dysfunction [33,111,112]. We also identified that residual expression of gp120 and virus could activate Panx-1 and Connexin-43 hemichannels in several cell types, further increasing ATP release during chronic HIV infection. More importantly, high levels of ATP in the serum of HIV-infected individuals can be used as a biomarker of cognitive impairment and chronic infection.

## 4. HIV and Panx-1 Interactions

The Pannexin family of proteins was first identified due to their sequence homology with gap junction proteins in invertebrates, innexins [113]. Pannexins share structural topology despite the lack of sequence homology with connexins [114]. Panx-1 is the most extensively studied and characterized [115]. The other two Pannexin family proteins, Panx-2 and Panx-3, have a more selective, tissue-specific expression profile relative to Panx-1 [113]. Each of the three Pannexins has four transmembrane domains, conferring two extracellular loops, a cytoplasmic loop, and a cytoplasmic C-terminus that varies in length among them [116]. The N-terminus is the most conserved, but they have divergent sequences from the C-terminus [117]. Panx-1 was thought to form a hexameric channel complex, but recent publications suggested that the multimeric conformation of Panx-1 into pannexin channels is in a heptameric state [118,119,120]. Earlier work suggested that Panx-1 could form cell-to-cell gap junctions between cells, much like their connexin counterparts [121,122]. The evidence against this is the need for glycosylation on the extracellular loop for membrane trafficking, which is thought to preclude the dimerization of two channels from forming intercellular gap junctions [123,124]. Functional analysis by scrape-loading also demonstrates the lack of gap junction formation between Panx-1s of neighboring cells [123]. However, when the N-linked glycosylation site is altered, preventing this post-translational modification, structural biology data suggests gap junction formation is possible [125]. One recent article argued that gap junction-forming innexins contain N-glycosylation sites at their extracellular loops and could form gap junction types of channels [126]. Using TC620 cells, an oligodendroglioma cell line, dye and electrical cell coupling were shown; the cell-to-cell coupling was reduced when Panx-1 was either knocked down by siRNA or inhibited by an inhibitory peptide [126]. However, Panx-1′s capacity to form gap junctions is still an open question.

Functionally, Panx-1 allows the movement of molecules up to 1 kDa across the plasma membrane [127]. The channels have affinity for anionic substrates, as demonstrated by single-channel patch-clamp electrophysiology [128]. They are in a constitutively closed state, but several stimuli are associated with the opening of the channel, which allows the efflux of ATP, prostaglandins, and several small molecules [115]. One of the first prominent roles identified was the release of ATP by Panx-1 as a “find-me” signal, which was the signaling event for early apoptotic activity [129]. The ATP secreted by a cell could then act as a chemoattractant for phagocytes to clear cellular debris of the apoptosed cell [130]. This constitutively open state of Panx-1 is associated with a caspase-3 or caspase-7 cleavage event but is also seen in Panx-1 truncation at the putative cleavage site [129]. When various C-terminal truncations were made, varying activities of Panx-1 were demonstrated, suggesting that the C-terminal tail plays an autoinhibitory function [131]. A study involving the autoinhibitory domain of Panx-1 also demonstrated that the interaction between this domain and the Panx-1 pore is not necessarily specific, as some C-terminal scramble mutants showed similar activity to the wildtype [132].

The induction of Panx-1 opening has several different mechanisms, such as changes in ion concentrations, mechanical stimulation, or post-translational modification [115]. The opening of Panx-1 was shown to occur upon potassium ion treatment, particularly when extracellular K^+^ concentrations reached 50–60 mM [133]. A publication showed that an increase in intracellular calcium ions within oocytes could also induce the opening of Panx-1 [134]. However, this activity was not seen within mammalian cells [135]. This was examined by administering varying concentrations of calcium ions, the alteration or depletion of intracellular calcium, and treatment with a phospholipase C inhibitor using patch-clamping [135]. Regarding mechanical stimulation, suction-based assays showed stretch activation of Panx-1 channels within oocytes [136]. Later studies within mammalian cells demonstrated that the mechanical induction of Panx-1 opening is inhibited by adenosine and cyclic adenosine monophosphate (cAMP) analogs [137]. Within the same article, this group also showed that T302 and S328, putative protein kinase A (PKA) phosphorylation sites, are essential for the stretch activation of Panx-1 due to the phosphomimetic mutants showing reduced activation from the same stimulus [137]. Work using rat astrocytes derived from the optic nerve also demonstrated the release of ATP through Panx-1 using mechanical means and in response to hypotonic swelling [138].

As mentioned above, apoptosis-dependent Panx-1 activation occurs by the cleavage of the C-terminal tail [129]. Concatemeric Panx-1 studies showed the stepwise activation of Panx-1 by the subsequent cleavage of the C-terminal tails by electrophysiology [118]. Ablation of the caspase cleavage site using a D376A/D379A mutant, termed PANX1-CR, showed the suppression of ATP release and Panx-1 C-terminal cleavage [129,139]. A single-cell dynamics model using Förster resonance energy transfer biosensors demonstrated that caspase-3 activity and decreased intracellular ATP level temporally coincided during apoptosis [139]. The glycosylation state also plays a role in the trafficking of Panx-1 to the plasma membrane as tunicamycin treatment, which prevents global protein glycosylation, showed a reduction in the membrane-localized Panx-1 [123]. Interestingly, a glycosylation-devoid mutant, N254Q, can still form functional channels in mammalian cells [123]. However, a similar study with zebrafish Panx-1 demonstrated reduced function but corroborated the reduced membrane localization in mammalian cells [123,140].

Other post-translational modifications, such as S-nitrosylation and phosphorylation, are known to modulate Panx-1 activity [141]. The addition of a nitric oxide (NO) donor, sodium nitroprusside, showed the inhibition of Panx-1 by a cyclic guanosine monophosphate (cGMP) and protein kinase G-dependent mechanism [142]. When S206, the predicted protein kinase G recognition site, was mutated to alanine, the inhibition of Panx-1 by sodium nitroprusside was reduced [142]. Treatment by other NO donors such as S-nitrosoglutathione or diethylamine NONOate also showed inhibitory effects on Panx-1 activity in Panx-1-transfected HEK293T cells [143]. The mutagenesis of C40 and C346, the proposed S-nitrosylation sites, reduced the inhibition of Panx-1 by NO, which was argued to be independent of cGMP signaling [143]. These articles suggested NO has an inhibitory effect on Panx-1 within HEK293 cells, albeit through different mechanisms; another article presented data within hippocampal neurons, which suggested that NO enhanced Panx-1 opening during oxygen–glucose deprivation [144]. In terms of HIV, NO has shown antiviral and proviral effects on viral replication [145]. Regardless of the effect of NO on HIV replication, multiple studies have shown it to be elevated within the serum of HIV-infected patients relative to uninfected or cART-adherent patients [146]. Overall, the data on HIV are poorly known, but Panx-1 channels are mostly in a closed state in uninfected cells, but upon primary infection or during chronic infection, the channel becomes open, enabling ATP to concentrate in the serum of HIV-infected individuals (Figure 1). Future studies will examine the mechanisms described in this section in HIV-infected cells.

## 5. Panx-1 and Purinergic Signaling Axis in HIV Infection

ATP has many functions inside and outside of cells. It operates as the main “energy currency” within cells, a substrate for nucleic acid synthesis, and a secondary messenger [147]. Extracellular ATP, secreted by cells by either vesicular secretion or through channels such as Panx-1, plays a role in signaling inflammation as a neurotransmitter and platelet activation [147]. Once released, ATP is a strong inflammatory molecule quickly degraded by soluble or membrane-associated ATPases such as ectonucleotide triphosphate diphosphohydrolase 1 (CD39) [148]. CD39 catabolizes its substrate into AMP upon binding to ATP or ADP, which can subsequently be catabolized into adenosine by CD73 [149]. Adenosine deaminase, either soluble or CD26-associated, converts the adenosine into inosine [150]. ATP and subsequent metabolites function through P1 or P2 purinergic receptors [151] (Figure 2). The four subtypes of P1 receptors, A_1_, A_2A_, A_2B_, and A_3_, selectively bind to adenosine [151]. P2 receptors, which are activated by both purines and pyrimidines, include ionotropic P2X (P2X_1–7_) and metabotropic P2Y (P2Y_1,2,4,6,11,12,13,14_) receptors [151].

In non-pathological conditions, the role of Panx-1 in the release of ATP is demonstrated upon the stimulation of various membrane receptors [152]. The release of ATP through Panx-1 contributes to T cell activation in an autocrine fashion [153]. This Panx-1-dependent release of ATP also occurs in response to the binding of chemokines to T cells to induce cellular polarization and a migratory phenotype [154]. Similarly, when HIV binds to a target cell through receptor/co-receptor and gp120 interactions, Panx-1 briefly opens, releasing ATP [155,156]. The ATP released then activates purinergic receptors, P2X_1_ and P2Y_2_, leading to HIV fusion and subsequent entry [157]. Signaling through P2Y_2_ in response to HIV infection showed an increase in Pyk2 Y402 phosphorylation, similar to work within cell lines that demonstrated the binding of gp120 to CCR5 [158]. Other work in macrophages demonstrated that the NLR family pyrin domain-containing 3 (NLRP3) activation impairs HIV entry by inhibiting F-actin remodeling [159]. In general, inhibiting F-actin remodeling with reagents such as latrunculin reduces viral infectivity and HIV entry [159]. The activation of P2Y_2_ during HIV entry stimulates the recruitment of an E3 ubiquitin ligase to promote the degradation of NLRP3, allowing a more permissive state for HIV entry [159]. Canonically, NLRP3 inflammasome activation is associated with Panx-1 release of ATP and functional coupling with P2X_4_ and P2X_7_ [160]. Signaling through ionotropic receptors leads to the cleavage and activation of caspase-1, which in turn cleaves pro-IL-1β into mature IL-1β [161]. For HIV infection, the downregulation of NLRP3 is advantageous for HIV, not only for entry as mentioned above, but also due to the reduction of HIV replication in the presence of recombinant IL-1β [162]. The binding of CCR5 to gp120 activated Gα_q_ to propagate a Rac-1-mediated actin cytoskeleton remodeling for fusion [158]. A dissenting article suggested that both inhibition of CD39 by polyoxotungstate-1 (POM-1) and the treatment of high levels of ATP during or before the infection of macrophages inhibited HIV infection [163]. However, some polyoxometalates have shown inhibitory effects on reverse transcriptase (RT) in vitro, so some inhibitory effects on HIV replication are likely due to the direct inhibition of RT [164].

Additionally, the treatment with high levels of ATP could also induce the internalization of Panx-1 within cell lines [165]. Further, the knockdown of either Panx-1 or P2Y_2_ showed a reduction in HIV infection and replication, suggesting that releasing ATP and transducing signals through P2Y_2_ are important for viral replication [155]. Lastly, the release of ATP and subsequent signaling requirements for HIV entry are evident when apyrase, which depletes extracellular ATP, is present during HIV infection in vitro [155].

Our data in simian immunodeficiency virus (SIV)-infected macaques indicate that blocking Panx-1 channel opening after SIV infection prevents immune compromise, leukocyte differentiation induced by the virus, transmigration into the CNS, and loss of complex synapses [33,110]. These data indicate that despite the complex animal model, mimetic peptides to Panx-1 can be used to prevent CNS damage and improve immune response. These data are outstanding because targeting these channels can provide an additional treatment to prevent immune and CNS damage and open several therapeutic avenues using purinergic/adenosine drugs to prevent and even reverse the damage produced by viral reservoirs in the current ART era.

Not only is Panx-1 required for HIV entry, but replication as well. At 48 to 72 hours post-HIV infection, Panx-1 becomes open in primary PBMCs and T cells, as demonstrated by dye uptake [156]. This is distinct from the entry-mediated opening or cytokine binding-like (such as SDF-1) as the opening of Panx-1 in these latter timepoints is prolonged [156]. This prolonged opening of Panx-1, similar to the entry-mediated opening, is also required for HIV replication, as the pharmacological inhibition or knockdown of Panx-1 after the infection has been established abolishes HIV replication [156]. Similar to entry, after infection is established, the pharmacological inhibition of P2X_1_, P2X_7_, and P2Y_1_ purinergic receptors reduces HIV replication within human macrophages [157]. Some evidence supports the purinergic receptor antagonist, oxidized ATP [146], which can suppress reverse transcriptase in vitro using either virus-free enzymes or a cell-free system with virus particles [166]. Another purinergic receptor antagonist, suramin, has also shown inhibitory effects on RT in vitro [167]. These effects and the ones referred to in the above paragraph regarding POM-1 are all off-target effects of the pharmacological purinergic receptor inhibitors that influence HIV [168]. Therefore, phenotypes seen by treatment with these inhibitors must be analyzed carefully and be conducted in tandem with siRNA knockdowns to ensure the phenotype is recapitulated.

## 6. Panx-1 and Purinergic Signaling Axis in NeuroHIV

Panx-1 has many roles within the CNS, such as neuronal development, neurite formation, dendritic spine development, and synaptic plasticity [169]. Knockout models of Panx-1 and pharmacological inhibition demonstrate aberrations in neuronal cell maintenance and neurite formation [170]. Aside from Panx-1’s role in normal physiological function and development, it has demonstrated multiple roles in pathological conditions [112]. A recent publication demonstrated that ATP within serum is a potential biomarker of neurocognitive impairment in HIV patients [110]. ATP plays many roles within serum, but, in excess, can be detrimental to anatomical barriers [171]. Mouse models have shown that, through purinergic receptors, ATP can induce vascular inflammation and promote atherosclerosis, as evidenced by atherosclerotic lesions among the endothelia [172]. The deficiency of P2X_4_ within an atherosclerotic mouse model showed reduced cytokine production, decreased leukocyte invasion within the endothelium, and reduced endothelial expression of adhesion molecules [173]. Multiple in vitro models have recapitulated ATP-induced blood–brain barrier [67] permeability, including one demonstrating P2X_7_ receptor dependence [110]. This mechanism of purinergic signaling-induced blood–brain barrier permeability showed that the downstream signaling of the P2X_7_ receptor, including the expression of IL-1β and subsequent metalloprotease expression (MMP), induced the tight junction disruption of endothelial cells [174]. This cascade is characterized by Panx-1 coupling with P2X_7_ to induce IL-1β release within macrophages and inflammasome activation in neuronal cells [175]. Multiple in vivo and in vitro models have shown MMP-induced BBB dysfunction by tight junction protein degradation [176].

Adenosine and P1 agonists have also demonstrated detrimental effects on the BBB [177]. Adenosine receptor agonists are known to disrupt the BBB [178]. In vitro, BBB co-culture assays using adenosine receptor agonists show an increase in endothelial cell permeability to both 10 kDa. Dextran also increased the transmigration of T lymphocytes [179]. In addition, the adenosine receptor agonist NECA (1-(6-amino-9H-purin-9-yl)-1-deoxy-N-ethyl-β-d-ribofuranuronamide) induced an increase in BBB permeability in a mouse model [180]. In the same study, Lexiscan, an agonist to adenosine receptor A_2A_, demonstrated increased BBB permeability [180]. Another study showing the treatment of Lexiscan on either an endothelial cell line or primary human endothelial cells shows the reduction of P-glycoprotein expression and function, as well as an increase in the expression of MMP-9 [181]. P-glycoprotein, a drug efflux pump expressed by endothelial cells of the BBB, can have both a proviral and antiviral activity [182]. The expression of P-glycoprotein in HIV-infected cells was increased to the benefit of HIV, as the accumulation of a nucleoside reverse transcriptase inhibitor was reduced relative to uninfected cells [183]. At the BBB, the lack of P-glycoprotein demonstrated the reduced capacity to eliminate drugs and other toxic metabolites from the brain [184]. Although this may alleviate the decreased delivery of ART into the brain, the function of P-glycoprotein is to be a sentinel to eliminate or prevent the accumulation of toxic substances within the brain [182]. These data only denote the early understanding of the stages of signaling and related toxicity mediated by viral reservoirs, Panx-1 channels, ATP, and purinergic receptors. The data also indicate that the chronic nature of HIV in the current ART era is different from other diseases and requires a significant effort to prevent the associated damage observed in at least 50% of the HIV-infected population.

## 7. Conclusions and Future Studies

Currently, the status of NeuroHIV in the ART era requires a significant re-evaluation of several mechanisms described early on in the AIDS pandemic, including the identification of viral reservoirs and their diversity, mechanisms of survival, latency, reactivation, and bystander toxicity. However, more important are the host proteins involved in these mechanisms that are probably different from those described for the typical HIV life cycle. Our data identified that Panx-1, a large ionic channel, mediates the release of ATP and inflammatory lipids by a mechanism of infection, but can be replication-independent. Circulating levels of ATP can be used as a biomarker of cognitive impairment in the HIV-infected population, and we believe that ATP contributes to BBB dysfunction. However, we believe other ionic channels are also involved in entry, infection, and replication, and the generation and stability of viral reservoirs. Thus, understanding the interactions between viral and host components could provide new avenues of treatment to prevent and even reverse CNS damage in the HIV-infected population.

## 8. Patents

The described work resulted in the US patent 17/476,910.

## Figures and Tables

**Figure 1 cells-11-02245-f001:**
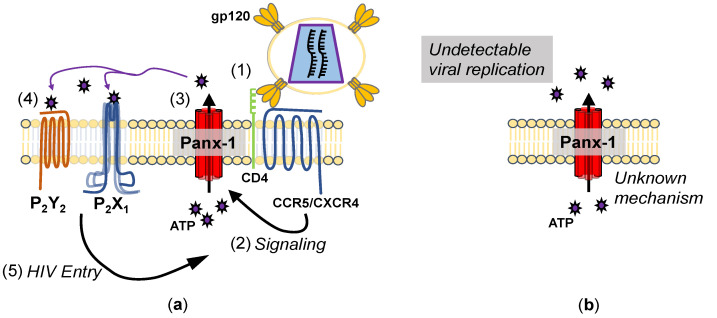
Mechanisms of Panx-1 opening during HIV infection. (**a**) Mechanism of Panx-1 opening during acute infection. (1) Upon binding to CD4 and CCR5 (or CXCR4), (2) signaling through this binding leads to Panx-1 becoming open, (3) releasing ATP. (4) The ATP released then binds to purinergic receptors, (5) allowing HIV entry. Inhibition of Panx-1 ATP release or P2 receptors inhibits HIV entry. (**b**) Mechanisms of Panx-1 opening during chronic infection. Peripheral blood mononuclear cells (PBMCs) isolated from HIV-infected patients show Panx-1 in a spontaneously open state, despite a lack of viral replication, but the mechanism is unknown.

**Figure 2 cells-11-02245-f002:**
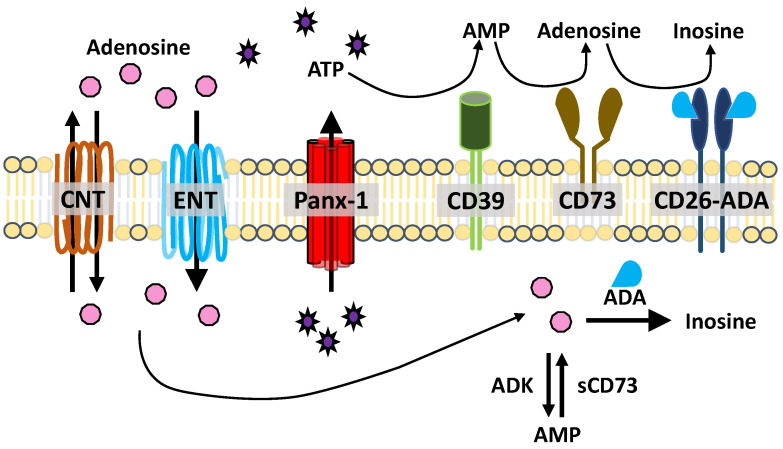
ATP release and metabolism. ATP is released through hemichannels, such as Panx-1. ATP (and ADP) are degraded into AMP by CD39. AMP is catabolized into adenosine by CD73. Adenosine is deaminated into inosine by adenosine deaminase (ADA), which is associated with CD26. Concentrative and equilibrated nucleoside transporters (CNT, ENT) facilitate nucleoside uptake and transport. Intracellular adenosine can be deaminated by ADA or converted to AMP by soluble adenosine kinase (ADK). Intracellular AMP is converted to adenosine by soluble CD73 (sCD73).

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
