# Peer review of "The Role of Pannexin-1 Channels in HIV and NeuroHIV Pathogenesis"

_cells, 2022, doi:10.3390/cells11142245_

Round 1
Reviewer 1 Report
Manuscript type: review
Journal Cells (ISSN 2073-4409)
Manuscript ID cells-1759426
Title The Role of Pannexin-1 Channels in HIV and NeuroHIV Pathogenesis
In this review manuscript, Dr. Hernandez Cristian, and Eugenin Eliseo provided a detailed overview of the mechanisms associated with ATP dysregulation and cognitive disease observed in the HIV-infected population. The ms is important as it provides important information on Pannexin-1 Channels in HIV and NeuroHIV Pathogenesis. The research topic is interesting and suitable for publication in cells. The manuscript has some minor issues that need to be solved. Please see below are some suggestions, which would allow the article to be improved for publication in Cells.
Comments and Suggestions for authors.
1. Acronyms should carefully be checked, and complete names should always be included when the protein/disease ect is mentioned for the first time. For instance, among many others, line 144, “HAND” should be “HIV-Associated Neurocognitive Disorder (HAND)” when mentioned for the first time. Please revise the entire text accordingly. I have found additional acronyms (see below)
2. The aim of the review sohuld be stated at the end of the introduction.
3. Line 29 ART should be anti-retroviral therapy (ART) when mentioned for the first time
4. Line 85 Among neurotropic viruses, authors can mention JC polyomavirus PMID: 25623836
5. Line 94 the fist sentence sohuld be below the sub-head title. Same considerations can be made at lines 112, 138-139, Moreover I suggest removing the / symbol for a better reading
6. Line 99 a unnecessary parenthesis should be removed
7. Line 124 IP3 should be mentioned as Inositol trisphosphate when mentioned for the first time in the text
8. Line 144 “HAND” should be “HIV-Associated Neurocognitive Disorder (HAND)” when mentioned for the first time
9. Lines297-299 Pannexin-1 channel also mediates NLRP3 inflammasome activation as a consequence of a a functional interplay between the channel and purinergic receptors P2X7 and P2X4 (please see https://www.mdpi.com/2072-6694/14/5/1116). This important information and supporting reference should be included
10. Figure 1 panel b is lacking in the lateral black border
11. Figure 2 quality should be improved.
12. Line 342 “in vitro” as wel as other Latinisms should be italic
13. Line 364 Meaning of BBB? blood-brain barrier?
Author Response
Manuscript type: review
Journal Cells (ISSN 2073-4409)
Manuscript ID cells-1759426
Title The Role of Pannexin-1 Channels in HIV and NeuroHIV Pathogenesis
In this review manuscript, Dr. Hernandez Cristian, and Eugenin Eliseo provided a detailed overview of the mechanisms associated with ATP dysregulation and cognitive disease observed in the HIV-infected population. The ms is important as it provides important information on Pannexin-1 Channels in HIV and NeuroHIV Pathogenesis. The research topic is interesting and suitable for publication in cells. The manuscript has some minor issues that need to be solved. Please see below are some suggestions, which would allow the article to be improved for publication in Cells.
Comments and Suggestions for authors.
- Acronyms should carefully be checked, and complete names should always be included when the protein/disease etc is mentioned for the first time. For instance, among many others, line 144, “HAND” should be “HIV-Associated Neurocognitive Disorder (HAND)” when mentioned for the first time. Please revise the entire text accordingly. I have found additional acronyms (see below)
Answer: We included all acronyms as requested .
- The aim of the review should be stated at the end of the introduction.
Answer: Excellent point, we added the purpose of the review as requested.
- Line 29 ART should be anti-retroviral therapy (ART) when mentioned for the first time
Answer: We fixed the issue. Thank you
- Line 85 Among neurotropic viruses, authors can mention JC polyomavirus PMID: 25623836
Answer: We added JC polyomavirus, as well as multiple other neurotropic viruses.
Line 94 the fist sentence should be below the sub-head title. Same considerations can be made at lines 112, 138-139, Moreover I suggest removing the / symbol for a better reading
Answer: We fixed the title headings and first line of the section texts. We would like to preserve the “microglia/macrophage” terminology due that most published analysis involve the use of CD68 or Iba-1 staining.
Line 99 a unnecessary parenthesis should be removed
Answer: Thank you.
Line 124 IP3 should be mentioned as Inositol trisphosphate when mentioned for the first time in the text
Answer: Fixed.
- Line 144 “HAND” should be “HIV-Associated Neurocognitive Disorder (HAND)” when mentioned for the first time
Answer: Thank you
- Lines297-299 Pannexin-1 channel also mediates NLRP3 inflammasome activation as a consequence of a a functional interplay between the channel and purinergic receptors P2X7 and P2X4 (please see https://www.mdpi.com/2072-6694/14/5/1116). This important information and supporting reference should be included
Answer: This is extremely relevant information to inflammatory signaling. We included both the functional interplay between these three factors, as well as included a relationship to HIV.
Figure 1 panel b is lacking in the lateral black border
Answer: This puzzled us as on the word document, it was intact but the print preview omitted the border. We redid the borders to address this.
- Figure 2 quality should be improved.
Answer: We included a higher resolution image for Figure 2.
- Line 342 “in vitro” as well as other Latinisms should be italic
Answer: All instances of in vivo and in vitro have been italicized within the text.
- Line 364 Meaning of BBB? blood-brain barrier?
Answer: We included the acronym within parentheses of the first mention of the blood-brain barrier.
To Reviewer 1
We would like to thank Reviewer 1 for their very careful and thoughtful revisions. We believe that their attention to detail has strengthened the manuscript in both content, as well as accuracy in scientific nomenclature
Reviewer 2 Report
Summary: This is a very nice review of the role of pannexin-1 in HIV with an emphasis on its potential contributions to neuroHIV. The review is timely as neuroHIV is still a large problem in ART-suppressed individuals. Why certain individuals have neuroHIV that shows as HIV-1-associated neurologic disorders (HAND) and others do not is still an open question. This group has contributed significantly to this field of literature and their work on ATP regulation in the CNS and pannexin-1 in this mechanism has led to the review of this concept in the current paper. Overall, the paper does a good job of framing the issue and discussing the mechanisms of pannexin-1. However, there are some issues in the abstract, introduction and first section on viral reservoirs that need to be addressed to link these sections to the main theme of the manuscript. See specific comments below.
Suggested critiques:
1. Abstract: The abstract is more focused on the role of ATP and PGE2 as biomarkers and underlying mechanism of neuroHIV. Given the title and the main concept of the manuscript, the authors need to link in pannexin-1 to the abstract. This reviewer might suggest that instead of focusing the abstract to ATP as a biomarker the authors link how pannexin-1 regulates ATP and underlies neuroHIV pathogenesis.
2. In the introduction (paragraph 2), the authors spend some time discussing “shock and kill” cure strategies. It is unclear how this section relates to the rest of the manuscript and the authors never link back to this concept. This reviewer would suggest removing this section or provide the reader some rationale as to why this relates to the rest of the paper.
3. In the introduction (paragraph 3), the authors spend some time discussing viral reservoirs. This is more focused on peripheral blood reservoirs like CD4 T cells. The authors then go on to spend section 2 reviewing CNS reservoirs. While both sections are scientifically accurate, why they are discussed and how they link to the main concept of the review (sections 3, 4, 5, and 6) are unclear. It is suggested to rework these sections to provide a rationale and link to the rest of the manuscript. Otherwise the review might benefit from removing these.
Author Response
Comments and Suggestions for Authors
Summary: This is a very nice review of the role of pannexin-1 in HIV with an emphasis on its potential contributions to neuroHIV. The review is timely as neuroHIV is still a large problem in ART-suppressed individuals. Why certain individuals have neuroHIV that shows as HIV-1-associated neurologic disorders (HAND) and others do not is still an open question. This group has contributed significantly to this field of literature and their work on ATP regulation in the CNS and pannexin-1 in this mechanism has led to the review of this concept in the current paper. Overall, the paper does a good job of framing the issue and discussing the mechanisms of pannexin-1. However, there are some issues in the abstract, introduction and first section on viral reservoirs that need to be addressed to link these sections to the main theme of the manuscript. See specific comments below.
Suggested critiques:
- Abstract: The abstract is more focused on the role of ATP and PGE2 as biomarkers and underlying mechanism of neuroHIV. Given the title and the main concept of the manuscript, the authors need to link in pannexin-1 to the abstract. This reviewer might suggest that instead of focusing the abstract to ATP as a biomarker the authors link how pannexin-1 regulates ATP and underlies neuroHIV pathogenesis.
Answer: We have changed the abstract to emphasize Pannexin-1 and its relationship to ATP and NeuroHIV.
- In the introduction (paragraph 2), the authors spend some time discussing “shock and kill” cure strategies. It is unclear how this section relates to the rest of the manuscript and the authors never link back to this concept. This reviewer would suggest removing this section or provide the reader some rationale as to why this relates to the rest of the paper.
Answer: We included the content about shock and kill to emphasize the nature of HIV infection, particularly that it is without a cure and is a chronic disease. We also expanded this section to focus into the viral reservoirs issue.
In the introduction (paragraph 3), the authors spend some time discussing viral reservoirs. This is more focused on peripheral blood reservoirs like CD4 T cells. The authors then go on to spend section 2 reviewing CNS reservoirs. While both sections are scientifically accurate, why they are discussed and how they link to the main concept of the review (sections 3, 4, 5, and 6) are unclear. It is suggested to rework these sections to provide a rationale and link to the rest of the manuscript. Otherwise the review might benefit from removing these.
Answer: We have expanded on this sections and better tight together the point of Pannexin-1 in viral reservoirs .
To Reviewer 2
We appreciate the comments by Reviewer 2 as we feel as though their comments were well received and yielded a more cohesive manuscript.
Round 2
Reviewer 2 Report
The responses have modified the manuscript and is ready for publication.